# OpenReview forum: "DynaBO: Dynamic Model Bayesian Optimization for Tokamak Control"
_ICLR.cc/2025/Conference — Submitted to ICLR 2025_

### Official Review · Reviewer_rNch · 2024-10-29

**Soundness:** 3
**Presentation:** 4
**Contribution:** 4
**Rating:** 8
**Confidence:** 3

**Summary:**

The manuscript describes an online optimization framework for avoiding so-called tearing-mode instabilities, when operating tokamak reactors. The manuscript is well-written, clear, and includes successful experimental evaluations (!) of the proposed optimization strategy.

The work has a great tutorial character and a real-world implementation/application flavor from which the machine learning and specifically the RL community would benefit. While the individual building blocks of the pipeline (recurrent neural networks for modelling dynamics; Gaussian processes; Bayesian optimization) are standard, the manuscript shines by presenting real-world experiments that are convincing and challenging to do. As a result, I suggest to accept the manuscript, as it would broaden the scope of work presented at ICLR and showcases an exciting and impactful example, where machine learning approaches solve challenging real-world problems related to dynamical systems.


Detailed comments/questions:
- the variable h seems to have been reused with different meanings. On p. 3 below (3), h maps to an electron cyclotron heating profile and on p. 4, h predicts the probability that a tearing mode occurs. -> Please update the manuscript accordingly.

- the gyrotron angles are kept fixed throughout each experiment. How limiting is this assumption? Have the authors considered relaxing this assumption, for example by introducing profiles that are computed a-priori? In reinforcement learning this is typically called the "options framework".

- how significantly does the online optimization improve performance in the practical experiments? The practical experiments include eight iterations, and it is unclear to me whether similar results could also have been obtained without the online updates? Is there a way to conduct experiments for a longer period of time or is this simply not doable on the DIII-D reactor?

- related to the last question: is there a more fine-grained performance metric rather than "Tearing Instability Avoided Yes/No"? This might help to underscore the effectiveness of the online optimization.

**Strengths:**

see above

**Weaknesses:**

see above

**Questions:**

see above

---

> ### Author Response · Authors · 2024-11-24
>
> Thank you for your comments. Here are the answers to your major questions:
>
> Options framework for gyrotron angles:
> Thank you for this suggestion. ECH dynamics are still poorly understood, and how to best deploy ECH for tearing instability control is still an open question. Past experiments indicate that varying ECH overt time increases the probability of causing plasma disruptions, which is undesirable because it reduces the potential for nuclear fusion. This motivates our steady-state setting. However, even in this setting, there is no established way of choosing ECH. After advances are made to solve this problem, we believe the options framework will become very interesting.
>
> Significance of online optimization:
> This is a very interesting question. Providing a clear answer is difficult, as this would entail rerunning the same experiments without updating the model and for a higher number of shots, which is difficult due to DIII-D-related constraints. To determine what to expect in our case with higher confidence, we have further analyzed historical data and found 61 experiments with very similar settings. On average, the historical data exhibits a rate of tearing instability of 77%, which is significantly higher than we obtained.
>
> Other performance metric:
> Thank you for this suggestion. We now also report how long we operate the tokamak without observing tearing instabilities. Our approach yields an average stable operation time of 3339 ms. The historical data with similar settings exhibits an average stable operation time of 2424 ms. For reference, the maximal duration of a full shot at DIII-D is approximately 7 s, the average is approximately 6 s.

---

> > ### Author Response · Authors · 2024-12-02
> >
> > Dear Reviewer rNch,
> >
> > Thank you again for your review. We are approaching the end of the discussion phase. If possible, please let us know if our responses address your concerns and if you have any additional questions, so we can include them in the revised version of the paper.

---

### Official Review · Reviewer_vpif · 2024-11-04

**Soundness:** 3
**Presentation:** 3
**Contribution:** 3
**Rating:** 5
**Confidence:** 4

**Summary:**

This paper presents a method for controlling tokamak fusion reactors with the objective of maximizing the time to an instability failure. Within each shot (experiment trial) an RPNN models the system dynamics at a fine timescale. Across shots, a GP refines the RPNN predictions by predicting the time-to-failure residual as a function of shot-level control variables. Experiments on historical data and live experiments show a significant advantage over baselines.

**Strengths:**

This is a high-profile and high-stakes application. The live experiments on a working fusion reactor are impressive. The experiment results apparently show an advance over SOTA in this area.

**Weaknesses:**

The work is light on theoretical advances. The RPNN and its particular encoding of states and actions appear not to be new (please correct if I'm wrong). The main contribution is to use a GP to learn residuals of this model.

A primary motivation given for the proposed method is to model changes in the hardware (e.g., aging and reconfiguration). The model is said to operate on two timescales. Thus I expected time to be included in the GP input space as a proxy for hardware changes. However, the GP is not time-indexed. As formulated here, the GP is exchangeable, meaning its posterior would be invariant to permutation of the training data. Thus it doesn't learn rapidly from new data as claimed, in the sense that recent data have greater influence than older data.

**Questions:**

For the live experiments sec 4.2 seems to suggest the GP was trained just for the 8 shots ("8 BO iterations") but sec A.1 seems to say it was first trained on 125 historical shots ("For online testing we use this as training set"). Please clarify.

Sec 2.2: It would help to state the shapes of variables, e.g. what is a scalar and what is a field (I know the details are in the appendix).

Sec 4.1: Why is there any variety in the action of the RPNN planner? Line 323 seems to indicate there is no exploration.

Line 201: mu should be eta

(7): sigma_n

There is inconsistency between a^g and a_g.

line 285: uncontrolled but not unknown because you measure the ECH profile

(10) Is tau the actual maximum in the dataset? Is t^(i)_TM the empirical time to failure for the planner's choice of a_g?

---

> ### Author Response · Authors · 2024-11-24
>
> Thank you kindly for your review. Here are our answers to your major comments:
>
> TIme-indexed GP:
> Thank you for this suggestion. We have conducted several additional experiments using a time-indexed GP and different kernels. Please refer to the table in the general comment above for a summary. The results are competitive, yet they are not better in all cases. One possible explanation is that the reliability of the data depends on multiple factors, many of which cannot be explained exclusively as a continuous function of time, e.g., particle absorption and release by the tokamak wall, and the presence of impurities due to previous experiments.
>
> Variety in the RPNN planner action:
> The RPNN does not explore, as it fully trusts its prediction. This results in the RPNN always picking the same ECH profile in each bin. In the presented results, we cluster multiple bins together for different intervals of $\beta_N$. Hence, each figure corresponds to multiple bins, hence different ECH picks obtained with the RPNN.

---

> > ### Author Response · Authors · 2024-12-02
> >
> > Dear Reviewer vpif,
> >
> > Thank you again for your review. We are approaching the end of the discussion phase. If possible, please let us know if our responses address your concerns and if you have any additional questions, so we can include them in the revised version of the paper.

---

> > > ### Comment · Reviewer_vpif · 2024-12-02
> > >
> > > Thank you for your replies and additions to the paper.
> > >
> > > The inclusion of the time-dependent kernel is very useful (as are the variants with zero mean and different kernel shapes), and it's interesting that time-dependence does not consistently help. Unfortunately this result goes against much of the motivation of the paper:
> > >
> > > * "combines a high-frequency data-driven dynamics model with a low-frequency Gaussian process" (018)
> > > * "time-dependent model changes cannot be neglected" (061-062)
> > > * "We model the system at two different time scales" (189)
> > > * "time-dependent fluctuations in the environment variables, e.g., due to maintenance or hardware changes" (224-225)
> > >
> > > These statements clearly suggest a dynamic model at both timescales: the RNN for within-shot dynamics and the GP for changes across shots. (The 2nd and 4th argue this is the type of model that is needed, and the 1st and 3rd imply this is what you actually did.)
> > >
> > > On rereading the paper I realized you may have merely meant the GP captures distribution shift between the data the RPNN was trained on and the current configuration of the apparatus (i.e., a simple model of time consisting of "past" and "present"). However, this doesn't square with the fact that the GP was conditioned on 125 historic shots in addition to the 8 live ones. All 133 (or 125+n for the predicting n+1st live shot) are treated exchangeably by the GP without time-dependent kernel.
> > >
> > > This all leads to the conclusion that the GP is simply learning residuals of the RPNN. This is a nice approach but I still don't think it's quite enough of a theoretical advance.

---

> > > > ### Author Response · Authors · 2024-12-02
> > > >
> > > > Thank you for these interesting observations.
> > > >
> > > > As you pointed out, the GP is modeling the RPNN residual. We will make this clear in the revised paper.
> > > >
> > > > The two timescales in comments 1 and 3 refer to timescales in a single shot: The RPNN takes the complete time-dependent actuator variables as input and makes a step-by-step prediction of the system dynamics, whereas the GP takes the ECH profile averaged over the full shot to make predictions.
> > > >
> > > > The time-dependent changes in environmental dynamics (comments 2 and 4) refer to a broad class of changes that occur over time and can affect the experiment differently. While some of these changes affect the experiment in a way that correlates with time, e.g., permanent hardware changes, some are more challenging to model, e.g., a changing number of gyrotrons due to hardware failures or changes to the plasma due to the introduction of impurities by previous experiments. Most of these changes are not captured by our RPNN and are generally difficult to model. This motivates using a GP, which aims to quickly learn the model discrepancy using exclusively trustworthy data.
> > > >
> > > > Regarding the 125 shots used to condition the GP for the live experiment, we ensured they faithfully matched the experimental setup by hand-picking them together with DIII-D scientists. We ensured these past experiments had no additional impurities and the hardware setup was similar. These strict requirements are reflected in the low number of shots used (125) compared to the total number of shots used to train the RPNN (15000). Although pre-conditioning on zero data was possible, this would have potentially resulted in an extended exploration phase, which would have been impractical due to the low number of allowed experiments. With this in mind, we believe this setup provided the best boost in terms of prior knowledge without corrupting the model too much with inaccuracies. We also note that we allow for corruption in the data due to the noise variance in the GP.
> > > >
> > > > We will include these discussions and clarify these points in the revised paper.

---

### Official Review · Reviewer_WVGE · 2024-11-04

**Soundness:** 3
**Presentation:** 2
**Contribution:** 3
**Rating:** 5
**Confidence:** 3

**Summary:**

This paper introduces a Bayesian optimization algorithm to find steady and open-loop control values for the gyrotron angles in a tokamak in order to prevent a certain type of plasma instability from happening. The method is applied to historical data from a tokamak as well as to live experiments in the same tokamak.

**Strengths:**

The paper provides a good introduction on the control of plasma instabilities in tokamaks. The proposed method is original in that it combines an RPNN with a GP for the objective function to optimize (here, the time to tearing instability). The RPNN, which provides a prior mean to the GP, is trained with historical data, while the GP is "trained" with live data acquired during the Bayesian optimization iterations.

**Weaknesses:**

1. It is not clear whether the authors claim to introduce a general framework for BO that uses a special prior mean, or whether they claim to introduce a specific method to control tearing instabilities in a tokamak. If it is the former (as the abstract suggests), the paper needs a much more comprehensive results section containing many more example systems and baseline methods. If it is the latter, the novelty from a purely ML standpoint seems limited and the work might be more appropriate in a field-specific venue.
2. The main novelty of the BO framework put forth in this paper seems to be the GP prior mean, computed using the RPNN trained on historical data. However, the authors clearly observe in Section 4.1 that "the RPNN is not accurate enough to faithfully predict tearing instabilities". Thus, I am not convinced that the GP prior mean in fact leads to faster or better convergence during the BO iterations. One critically missing baseline in Section 4 is to use the same BO algorithm but with zero mean.
3. In the introduction, the authors explain in detail the shortcoming of model-free and model-based RL methods when applied to controlling plasma dynamics, before presenting BO as a potentially superior method. This sounds extremely misleading to me as BO is only applicable to finding open-loop (and, in this case, steady) control actions, as opposed to the closed-loop and time-dependent control actions that RL policies typically output. In general, I feel that this paper is making large simplifying assumptions when it comes to the plasma control setting (open-loop and steady gyrotrons angles during an experiment, ability to control the ECH profile directly, etc), and it would be helpful to list all these simplifications at the outset in Section 2, rather than dispersed throughout Section 3.

**Questions:**

1. The authors mention in Section 2.1 that "Electron Cyclotron Heating (ECH) is one of the most effective methods to counteract tearing instability". This begs the question: does ECH already work, and if yes, what kind of control algorithm is currently used?
2. In Section 4, it is mentioned that the RPNN is "trained using 15,000 one-step state transition observations". Is the RPNN trained using only one-step trajectories?
3. Overall, I found Section 4.1 very difficult to understand. For example, what is meant in lines 321 to 323? In the right column of Figure 5, why are most of the curves only red?
4. In the live experiments, did each of the 8 experiments involve 8 BO iterations (and therefore 8 separate runs)?
5. Many typos throughout the paper: sometimes the bar is missing above $\beta_N$ for the target pressure, wrong index in line 228, the $\arg \min_t T_t$ formulation in equation (6) should simply be $\min t$, incomplete sentence in line 279, etc.

---

> ### Author Response · Authors · 2024-11-24
>
> Thank you kindly for your comments and questions. Here are the answers to your major points:
>
> Weaknesses 1 and 2 (general vs specific framework + additional experiments):
> Thank you for these observations. We have conducted extensive additional experiments on the historical dataset (please see the table in the general response above for a summary): we now compare our approach to a zero-mean GP and time-conditioned DynaBO (our approach with a time-conditioned GP). We also conduct experiments with different kernel structures and hyperparameters. Our approach consistently outperforms the zero-mean GP except when the kernel is linear, which is essentially misspecified, leading to no convergence in cumulative regret for any setting. The approach with time-conditioning is competitive, and we have included it in the revised manuscript, where we thoroughly discuss the results.
>
> Weakness 3 (summary of assumptions):
> Thank you for pointing this out to us. In the revised paper, we will summarize the assumptions required by our model at the beginning of Section 3. Please note that although we search for actions ECH space, we allow for limited controllability of ECH: After our method picks an ECH profile, we still measure the executed ECH and use only that measurement to update our model, not the desired query. This allows for discrepancies between the desired query and the true ECH profile. We have included this clarification in the revised paper.
>
> Question 1 (SOTA for ECH control):
> Although ECH has shown promise in controlling tearing instabilities, the dynamics behind this method are still poorly understood. Past experiments indicate that it is difficult to change ECH over time without causing a significant disruption in plasma performance. Hence, most experiments aim to find an appropriate steady-state regime. As of now, there is no consensus on what the best configuration is. In many DIII-D experiments that use ECH to control tearing modes, the gyrotron angles were chosen manually based on subjective evaluation of past experiments. We clarify this point in the revised paper.
>
> Question 2 (RPNN training):
> The sentence refers to 15,000 shots (rollouts) of approximately 5 seconds each. We clarify this in the revised paper.
>
> Question 3 (Clarity of Section 4.1):
> We have made several improvements to Section 4.1. We have streamlined the presentation of the results and revised the text for clarity. We have also improved the figures, which now depict cumulative regret.
>
> Question 4 (BO iterations in live experiments):
> Each experiment corresponds to a single BO iteration. During each experiment, the tokamak is operated for approximately 5 seconds. We have clarified this in the paper.

---

> > ### Comment · Reviewer_WVGE · 2024-11-27
> >
> > I thank the authors for answering my questions and updating their paper. The added clarifications throughout the paper, the comparison with the zero-mean GP, and the revised figure 2 are notable improvements. That being said, the paper still generally leaves an impression to be "unfinished": many paragraphs in Section 3 and 4 are poorly written, there was a last-minute revision of the key live experiment results, and the first weakness that I noted above has not been addressed. Taking everything into consideration, I will update my score to 5.

---

### Official Review · Reviewer_DxoJ · 2024-11-04

**Soundness:** 3
**Presentation:** 3
**Contribution:** 2
**Rating:** 6
**Confidence:** 3

**Summary:**

This paper discusses  a  Bayesian optimization approach to improve data efficiency and adaptability in control problems with limited, evolving data in the domain of nuclear fusion.

**Strengths:**

- Very interesting domain
- The paper is written rather well and easy to understand
- Overall the approach to use a high-frequency model (here, an RNN) and a low frequency model (here a GP) makes sense
- Real empirical evaluations on a Tokamak

**Weaknesses:**

- Under normal circumstances the proposed method would struggle, because it does not plan and only does 1-step planning so to say. The authors get around this problem by employing models of different frequency (RNN and GP) and by infusing domain knowledge into the architecture (page 5). While reasonable, this limits the generality of the approach.

- Experiments limited:
1. In the introduction the authors say: "In the case of tokamak control, BO has been used, e.g., to control the rampdown of a real tokamak (Mehta et al., 2024), and to control neutral beams in a tokamak simulator (Char
et al., 2019). However, the work of Mehta et al. (2024) does not address critical plasma instabilities, whereas Char et al. (2019) relies on a simulator. Moreover, these methods use a poorly specified prior and require an extensive amount of experiments to perform well."

These shortcomings could have been evaluated empirically in this paper.  E.g. one could have shown empirically that "Moreover, these methods use a poorly specified prior and require an extensive amount of experiments to perform well."

2.  While I acknowledge that for experiments in 4.2 comparisons and extensive studies will be difficult to do due to external constraints for 4.1 additional analyses would have been needed. Just a couple of proposals:
 How robust is the proposed method (compare different choices of kernel functions, RNN hyper-parameters,  available data, alternative modeling methods?)



Minor:
(Line 221, page 5): \cittep

**Questions:**

n/a

---

> ### Author Response · Authors · 2024-11-24
>
> Thank you for your comments. We have addressed all major points as detailed below:
>
> Weakness 1 (Empirical evaluation of Mehta et al. (2024) and Char et al. (2019)):
> Thank you kindly for this suggestion. Both the method of Char et al. (2019) and Mehta et al. (2024) use a prior mean of zero, which we have considered in additional experiments (see general response above). Furthermore, the results reported in their papers consider a considerably higher number of experiments than the ones in our paper (175 in Char et al. (2019) and 45 in Mehta et al. (2024)). Please note that a one-to-one comparison is not possible due to the different nature of the tasks.
>
> Weakness 2 (Robustness of proposed method):
> To address this comment, we have included several additional experiments. These include varying the hyperparameters, using different GP kernels (linear, Matern, and Gaussian), substituting the RPNN prior mean with a zero prior mean, and conditioning the GP on time (please see the table above for a summary). Please note that we did not consider changes to the RPNN, since a similar study has already been done in Char et al. (2024), which concluded that RPNNs have good properties for a similar setting.
>
>
> I Char, Y Chung, W Neiswanger, K Kandasamy… - Advances in Neural Information Processing Systems, 2019
>
> V Mehta, J Barr, J Abbate, MD Boyer, I CharAutomated experimental design of safe rampdowns via probabilistic machine learning  - Nuclear Fusion, 2024
>
> I Char, Y Chung, J Abbate, E Kolemen, J Schneider - arXiv preprint arXiv:2404.12416, 2024

---

> > ### Author Response · Authors · 2024-12-02
> >
> > Dear Reviewer DxoJ,
> >
> > Thank you again for your review. We are approaching the end of the discussion phase. If possible, please let us know if our responses address your concerns and if you have any additional questions, so we can include them in the revised version of the paper.

---

### Author Response · Authors · 2024-11-24

Thank you kindly to all reviewers for their thorough and helpful feedback. We have conducted several additional experiments on the historical data and compared it to two additional baselines using various different kernels. The cumulative regret after 500 iterations can be seen in the table below.


|                        | RBF  | $l=0.01$  $\hspace{.3cm}$     | RBF  |$l=0.1$        $\hspace{.3cm}$         | RBF  |$l=1$           $\hspace{.3cm}$        | Matern  |$\nu=0.5$        $\hspace{.3cm}$   | Matern  |$\nu=2.5$        $\hspace{.3cm}$    | Linear         |              |
|------------------------|-------------------|--------------|------------------|--------------|------------------|--------------|---------------------|--------------|---------------------|--------------|------------------|--------------|
|                        | Mean              | Std. Err     | Mean             | Std. Err     | Mean             | Std. Err     | Mean                | Std. Err     | Mean                | Std. Err     | Mean            | Std. Err     |
| DynaBO                 | 10.553            | 3.450        | 9.864            | 3.623        | 10.726           | 1.662        | 9.414               | 1.826        | 10.223             | 1.789        | 52.427         | 4.962        |
| DynaBO + Time          | 16.537            | 4.084        | 9.475            | 3.497        | 9.196            | 2.039        | 9.310               | 3.149        | 9.963              | 3.208        | 82.231         | 5.582        |
| Zero Prior   | 27.118            | 13.389       | 27.167           | 13.371       | 30.252           | 12.264       | 29.707             | 12.454       | 29.742             | 12.427       | 51.927         | 4.978        |

In the new paper, we also report the time of stable tokamak operation for the live experiments and the statistics for a more comprehensive set of historical experiments with very similar configurations. We have highlighted all significant changes in blue.

After internal discussions with DIII-D scientists, we concluded that we initially misrepresented two tearing mode occurrences that were difficult to detect. This results in a corrected tearing mode occurrence of 50%. We note that this is still better than the historical rates and that the stable operation time is also considerably improved, as some tearing modes occurred comparatively late in the experiment. In the revised paper, we now also report the full shot logs.

---

### Meta-Review · Area_Chair_JA76 · 2024-12-23

**Metareview:**

The authors propose a Bayesian Optimization (BO) approach which combines a high-frequency RNN model and a low-frequency GP model, such that the dynamics of the setting are captured well while the overall process can adapt rapidly to new data points. The key motivation (and evaluation framework) for this method comes from solving a key issue in an challenging real-world control problem, namely maximizing the time to an instability failure in a tokamak plasma.

The paper is generally well written, although a few paragraphs need further work to feel more “complete”. The application domain is particularly important and the application of the method to that domain is impressive. The authors apply the method  to historical data from a tokamak but also to live experiments.

The application of the method to a real, important application is the basis for the arguments put forward by the reviewer who championed the paper. However, at the same time other reviewers have made the point that in its current form, the manuscript doesn’t fully “do justice” to this important research, because it is not clear whether this is about modeling/controlling tokamak instabilities or about a general BO framework. Currently, this unclear motivation creates gaps in the storyline, and leaves the reader unconvinced.

Indeed, there are various “weaknesses” mentioned by the reviewers which I believe are caused by the above gaps. For example, as a more general BO framework, one would expect more extended evaluations (after the rebuttal period some comparisons were added but these were largely variations of the previous ones) or theoretical justification of the model. Regarding the latter, the reviewers point out certain potential theoretical limitations such as 1-step planning, or question whether dynamics are actually present in the timescales assumed by the authors.

Overall, it is rare to see nowadays a BO paper applied in such an important and challenging real-world scenario. Therefore, I expect to see this paper accepted in the future once the authors improve the storyline of the paper (boosting the corresponding parts as appropriate with further explanations and experiments)

**Additional Comments On Reviewer Discussion:**

The authors have clarified various parts of the text and provided an updated manuscript.
They also provided some new experiments (suggested by the reviewers), including variations of the model and comparison with a zero-mean mode and a time-dependent kernel.

The reviewers welcomed the above additions and the interesting discussion. However, they still remained overall unconvinced - details of these discussions have been included in the meta-review above as they were relevant to explain the final decision.

---

### Decision · Program_Chairs · 2025-01-22

Reject